# Understanding Diagnostic Costs Using Hospital-Based Encounters in the Year Before Diagnosis for Canadian Patients with Malignant Central Nervous System Tumours Compared to Common Cancers

**DOI:** 10.3390/curroncol32020096

**Published:** 2025-02-09

**Authors:** Linwan Xu, Keyun Zhou, Yan Yuan, Emily V. Walker

**Affiliations:** 1School of Public Health, University of Alberta, Edmonton, AB T6G 1C9, Canada; linwan2@ualberta.ca (L.X.); keyun1@ualberta.ca (K.Z.); yyuan@ualberta.ca (Y.Y.); 2Precision Analytics, Cancer Research & Analytics, Cancer Care Alberta, Alberta Health Services, Edmonton, AB T5J 3C6, Canada

**Keywords:** Canada, central nervous system cancer, colorectal cancer, diagnostic cost analysis, leukemia, lung cancer, quantile regression

## Abstract

Rare cancers pose significant diagnostic challenges, leading to more tests and higher healthcare expenditures (HEs). Understanding the financial implications of diagnosing rare cancers is crucial, particularly in Canada, where overall HEs are high (12% of the GDP in 2023). We investigated the pre-diagnostic hospital–based HE for patients with malignant central nervous system (CNS) tumours and compared it to patients with common cancers across Canadian provinces, using in-patient and ambulatory care data (2010–2014) from the Canadian Institute for Health Information. Pre-diagnostic HE was calculated as the change in total HE (in-patient and out-patient) during the 12 months before diagnosis, calculated as the HE within this period minus the average annual HE estimated over the two preceding years. Comparison groups included pediatric patients diagnosed with leukemia and patients aged over 15 diagnosed with colorectal cancer and lung cancer. We used quantile regression to estimate the adjusted effect of diagnosis with a CNS tumour on pre-diagnostic HE. The results indicated that HE for CNS patients was higher compared to those with common cancers. The top three factors contributing to HE variation were encounter type (in-patient/out-patient), province (Alberta/Ontario), and comorbidities (yes/no). Further investigation is warranted to understand the drivers of the cost differences.

## 1. Introduction

There is no universal definition for rare cancer [1,2]. In the European Union, a rare cancer is defined as one that affects fewer than 6 people per 100,000 annually [1]. In the United States, the threshold is fewer than 15 cases per 100,000 people annually [2]. While individually rare, these cancers collectively account for a substantial portion of all cancer diagnoses. Walker et al. [3] estimated that in Canada, 17.0% and 34.4% of annual cancer diagnoses were rare based on EU and U.S. definitions, respectively. Diagnosing rare cancers can pose a challenge due to their low incidence, which may lead to limited clinical experience in primary care settings. Additionally, the constellation of symptoms in the early stages may be non-specific and resemble those of more common conditions, which may contribute to misdiagnosis or delays in diagnosis [4]. Timely diagnosis is critical, as delays can negatively impact survival rates and lead to higher healthcare costs pre- and post-diagnosis [4]. Investigating the costs of rare cancers compared to common cancers is essential for understanding the financial burden these cancers impose on healthcare systems and for identifying the factors contributing to these cost differences.

Primary central nervous system (CNS) tumours account for approximately 2% of all annual cancer diagnoses in Canada, with around 36% of these being malignant [5]. This group of rare tumours is characterized by a wide range of histopathological subtypes, each associated with different treatment requirements and prognoses, making accurate classification essential for optimal patient management [5]. The classification system for CNS tumours has evolved over time, with updates from the World Health Organization (WHO) incorporating advancements in molecular and genetic research [6,7,8]. These updates have improved diagnostic precision, facilitated targeted therapies, and enhanced prognostic evaluations, enabling more personalized treatment plans and better overall management of patients’ conditions [7]. However, the advanced molecular classifications of these tumours further complicate diagnosis. As a result, the diagnostic interval for CNS tumours can be extended not only by the time required to rule out more common explanations for early symptoms but also by the need for specialized diagnostic tests and the expertise necessary to select the appropriate tests [4]. Given these challenges, the cost of diagnosing CNS tumours is likely higher than that of common cancers. To investigate this, we analyzed pre-diagnostic hospital-based healthcare expenditures (HEs) from 2010 to 2014 among patients with malignant CNS tumours in Canada and compared these costs to those for patients with common cancers across various provinces.

## 2. Methods

### 2.1. Data Coverage and Datasets

The data were provided by the Canadian Institute for Health Information (CIHI) from two databases: the National Ambulatory Care Reporting System (NACRS) and the Discharge Abstract Database (DAD). The NACRS covers out-patient care, including day surgery, community clinics, and emergency department visits. The DAD includes in-patient care and day surgery. Data coverage and organization can vary between provinces. Notably, out-patient clinic data often have low coverage in many provinces, and day surgery data are included in the NACRS database only for Ontario, Alberta, Prince Edward Island, and New Brunswick; for other provinces, these data were recorded in the DAD [9]. For our analysis, we focused on patients from Alberta and Ontario (ABON) because these provinces have the most complete datasets, which covered patient data on in-patient care, emergency department visits, community clinic visits, and day surgeries.

### 2.2. Patient Selection

The International Classification of Diseases, 10th Revision, Canada (ICD-10-CA) [10], diagnosis codes were used to identify patients diagnosed with malignant CNS tumours during the fiscal years 2010 to 2014, with each fiscal year covering the period from April 1 of this year to March 31 of the following year [9,11,12]. Common cancers include colorectal cancer (CRC), lung cancer (LC), and leukemia, as CRC and LC are the most frequently diagnosed cancers affecting both sexes [13], and leukemia is the most frequently diagnosed cancer in the pediatric population [14]. Multiple comparison groups were chosen to contrast CNS tumour diagnoses with cancers typically identified at earlier stages in primary care settings (e.g., CRC) and those commonly diagnosed at later stages in secondary care settings (e.g., LC). Given that CRC and LC are rare in pediatric populations [15,16], we compared costs for CNS patients with those of CRC or LC patients among adolescent and young adults (AYA) and older patients. Additionally, we compared costs for pediatric CNS patients with pediatric patients diagnosed with leukemia (Figure 1). Each unique patient ID obtained from the combined 2010–2014 data allowed us to extract all linkable historical patient information, including both female and male patients from ABON while excluding records of stillbirths and newborns.

### 2.3. Pre-Diagnostic Healthcare Expenditure Calculation

The diagnostic index date was defined as the earliest date on which a cancer diagnosis appeared in a patient’s record. From this index date, we traced back one to three fiscal years to extract all linkable historical records and estimate the cumulative pre-diagnostic costs for patients diagnosed with the cancers of interest each year. Pre-diagnostic healthcare expenditures (HEs) were calculated as the change in total HE during the year before cancer diagnosis, determined by subtracting the average annual HE from the two years preceding the diagnosis from the HE in the year prior to diagnosis (Figure 2). All cost estimates are in Canadian dollars.

### 2.4. Data Analysis

Change in total HE was the dependent variable; the investigated factors that may influence HE included cancer type, province, sex, age, encounter type, and comorbidity status. Age was categorized into four groups: 0–14 years (pediatric), 15–39 years (adolescent and young adults), 40–64 years (adults), and 65 years and above (seniors) [17]. The types of hospital encounters were in-patient and out-patient, with day surgery data included in the out-patient category. Comorbidity status was defined as either with or without comorbidity based on the comorbidities included in the Charlson Comorbidity Index [18,19]. We performed the Shapiro–Wilk W test for the estimated total diagnostic cost, and due to the highly skewed nature of the cost data, we used quantile regression analysis at the 50th and 90th percentiles to examine the associations between the dependent and independent variables without assuming normality or homoscedasticity [20]. The quantile regression is robust for outliers and allows the examination of the full cost distribution by different quantiles, capturing heterogeneity. In this study, the HE of patients in the 0–14 age group with CNS tumours was compared to those of pediatric patients with leukemia. For CNS patients in the other age groups, HE was compared to patients with CRC or LC, as these cancers are less common in the pediatric group [15,16]. The reference categories included patients with CNS cancer, those from Ontario, female sex, the 65+ age group, diagnosed under in-patient care, and patients with no comorbidity. Analyses were conducted using STATA version 17 (Stata Corp LLC, College Station, TX, USA), and figures were generated using the statistical software R 4.3.2 (R Core Team, Vienna, Austria).

## 3. Results

### 3.1. Cost of Patients with CNS vs. CRC/LC

#### 3.1.1. Patient Characteristics

The distributions of patient characteristics and pre-diagnostic HEs are shown in Table 1. From the cancer type perspective, CNS patients consistently had higher median and 90th percentile HE across most patient characteristics compared to CRC and LC patients. Overall, the median cost for CNS patients exceeded CRC patients by CAD 4.7K and LC patients by CAD 3.1K. This pattern persisted at the 90th percentile, with CNS patient costs being CAD 5.7K higher than CRC and CAD 5.3K higher than LC patients. From the provincial perspective, HE was generally higher in Alberta than in Ontario for all cancer types. In Alberta (AB), CNS patients cost CAD 1.7K more than CRC and CAD 0.1K more than LC patients at the 90th percentile, while in Ontario (ON), the differences reached CAD 6.4K. In general, costs between males and females were comparable. About 45% to 49% of the patients were female depending on cancer type. At the 90th percentile, female CNS patients cost about CAD 4.0K more than female CRC and LC patients. For male CNS patients, the cost difference was CAD 7.2K compared to CRC and CAD 6.3K compared to LC. From the age group perspective, most patients were aged ≥40 (81% of CNS, 98% of CRC, and 99% of LC patients). At the 90th percentile, CNS patients aged 40–64 had higher HE compared to other cancer types: CAD 8.6K more than CRC and CAD 6.2K more than LC patients. For those over 65, CNS patients had costs that were CAD 5.7K higher than CRC and CAD 7.1K higher than LC patients. Most CRC (66%) and LC patients (62%) were diagnosed through out-patient care, while only 45% of CNS patients were first diagnosed from this encounter type. Comorbidities prior to diagnosis were present in 7% of CNS, 12% of CRC, and 18% of LC patients. 

#### 3.1.2. Quantile Analysis

We included variables and scientifically relevant interactions in the final regression model if their *p*-value was below 0.2 at either the 50th or 90th percentile. For a 65+ year-old patient from Ontario with a CNS tumour, treated through in-patient care and without comorbidities, estimated costs were CAD 11.4K (95% CI: CAD 11.2 to 11.6K) at the 50th percentile and CAD 34.4K (95% CI: CAD 33.0 to 35.7K) at the 90th percentile. Cancer type, province, age, encounter type, and comorbidity status significantly impacted healthcare costs (Table 2).

When comparing the cost differences between patients with CNS cancer and those with CRC or LC, stratified by province and encounter type (Figure 3a), CNS cancer patients had significantly higher in-patient HE than CRC patients at the 50th percentile in both provinces. However, CNS cancer patients had significantly lower out-patient HE compared to LC patients, despite their in-patient HE being higher than LC patients in ON. At the 90th percentile, the HE differences between CNS cancer patients and common cancers became more pronounced (Figure 3b). CNS patients had significantly higher out-patient HE compared to CRC patients in both AB (CAD 3.2K [95% CI: CAD 1.0 to 5.4K]) and ON (CAD 4.2K [95% CI: CAD 2.8 to 5.7K]), and higher costs than LC patients in ON (CAD 2.0K [95% CI: CAD 0.5 to 3.5K]). Additionally, CNS cancer patients incurred higher in-patient HE than LC patients in both provinces (AB: CAD 2.9K [95% CI: CAD 0.5 to 5.3K]; ON: CAD 4.3K [95% CI: CAD 2.9 to 5.7K]). These findings suggest that CNS cancer patients tend to have higher HE than common cancer patients, especially at higher percentiles.

When comparing HE across provinces by cancer type and age group, HEs in AB were generally higher than in ON. At the 50th percentile (Figure 4a), HEs in AB were significantly higher than in ON for all cancer types, types of care, and age groups, except for CNS cancer patients aged 15–39 under out-patient care. The largest differences were observed for LC patients, with in-patient care costs in AB CAD 5.4 to CAD 5.5K higher than in ON and out-patient costs CAD 1.1 to CAD 1.2K higher. For CRC patients, in-patient care costs in AB were CAD 4.8 to CAD 4.9K higher than in ON, while out-patient care differences ranged from CAD 0.5 to CAD 0.6K. CNS cancer patients had in-patient cost differences of CAD 4.7 to CAD 4.8K and out-patient differences of CAD 0.4 to CAD 0.5K. At the 90th percentile (Figure 4b), HE in AB remained higher for in-patient and out-patient care for LC and CRC patients aged 40 and older. For CNS patients, HE in AB was higher across all age groups diagnosed through in-patient care, and for those aged 65 and older diagnosed through out-patient care.

### 3.2. Cost of Pediatric Patients with CNS vs. Leukemia

#### 3.2.1. Patient Characteristics

In our ABON pediatric cohort, 402 CNS cancer and 918 leukemia patients were included (Table 3). Overall, CNS cancer patients had higher HE than leukemia patients across all characteristics. Specifically, CNS cancer patients had CAD 4.5K and CAD 29.8K higher HE at the 50th and 90th percentiles, respectively. Geographically, approximately 65% of patients were from ON. In AB, CNS cancer patients cost CAD 2.9K and CAD 39.8K more than leukemia patients at the 50th and 90th percentiles, respectively, while in ON, CNS cancer patients had HE that exceeded leukemia patients by CAD 5.1K at the 50th percentile and CAD 12.5K at the 90th percentile. Additionally, CNS cancer patients in AB incurred CAD 26.8K more in HE than those in ON at the 90th percentile, while leukemia patient costs were comparable between the two provinces. More male patients (56% for both cancers) were included in the cohort. Female CNS cancer patients had median costs that were CAD 4.6K higher and CAD 21.6K higher at the 90th percentile compared to leukemia patients. Male CNS patients showed even larger differences, costing CAD 3.5K and CAD 45.0K more at the median and 90th percentile, respectively. Among CNS cancer patients, males had CAD 20.9K higher HE at the 90th percentile compared to females, whereas HE for leukemia patients was similar across sexes. A higher proportion of leukemia patients (72%) were diagnosed through out-patient care, compared to 46% of CNS cancer patients. Comorbidities were present in 2% of CNS cancer and 1% of leukemia patients prior to cancer diagnosis.

#### 3.2.2. Quantile Analysis

We included variables and interactions in the regression model where the corresponding *p*-values were below 0.2 at either the 50th or 90th percentiles. Due to a small sample size, comorbidity status was not included in the model. For a pediatric patient from Ontario with a CNS tumour, diagnosed through in-patient care, the estimated cost was CAD 14.3K (95% CI: CAD 13.2 to CAD 15.5K) at the 50th percentile and CAD 68.5K (95% CI: CAD 55.9 to CAD 81.0K) at the 90th percentile. Healthcare costs were significantly impacted by cancer type, province, and encounter type (Table 4).

When comparing the cost differences between patients with CNS cancer and those with leukemia, stratified by province and encounter type, at the 50th percentile, in-patient HEs were significantly lower for CNS cancer patients compared to leukemia patients in both AB and ON (Figure 5a). The estimated difference in costs was CAD 6.9K (95% CI: CAD 4.7 to 9.0K) in AB and CAD 7.7K (95% CI: CAD 6.2 to 9.2K) in ON. No significant differences were observed in out-patient HE in either province. However, at the 90th percentile (Figure 5b), CNS cancer patients had substantially higher in-patient costs in AB, with a notable increase of around CAD 29.3K (95% CI: CAD 5.3 to 53.2K) compared to leukemia patients. In contrast, no significant difference in HE was observed in ON at this percentile.

Significant cost variations were also observed between provinces. Among leukemia patients, in-patient HE at the 50th percentile was about CAD 8.7K (95% CI: CAD 6.7 to 10.7K) higher in AB compared to ON. For CNS cancer patients, in-patient care costs were markedly higher in AB, with differences reaching nearly CAD 9.6K (95% CI: CAD 7.5 to 11.6K) at the 50th percentile and CAD 37.7K (95% CI: CAD 15.1 to 60.3K) at the 90th percentile (Figure 6a). Additionally, CNS cancer patients in AB faced significantly higher out-patient HE at the 90th percentile, with a difference of CAD 25.2K (95% CI: CAD 3.9 to 46.6K) compared to ON (Figure 6b).

## 4. Discussion

This study used the quantile regression approach to compare pre-diagnostic HE between patients with CNS cancers and those with common cancers, specifically CRC, LC, and leukemia, in Alberta and Ontario. Our results show that CNS tumour diagnoses tend to be more costly, particularly at the 90th percentile. Generally, CNS patients had significantly higher costs than CRC patients in both provinces, with the exception of out-patient care at the 50th percentile and in-patient care at the 90th percentile, where no significant differences were observed. In comparison with LC patients, CNS patients had higher overall costs, except for out-patient care at the 50th percentile, where LC patients incurred greater costs. When comparing pediatric patients, leukemia patients had higher in-patient HE at the 50th percentile in both provinces, while CNS patients in Alberta incurred significantly higher in-patient HE at the 90th percentile. Additionally, we found that HE in Alberta was generally higher than in Ontario, particularly for in-patient care.

Although the economic burden of cancer has been a topic of research interest, a review of the literature did not yield many papers summarizing direct comparisons of the healthcare costs of rare cancers like CNS to common cancers. The findings from an analysis by De Oliveira et al. [21] were consistent with the results of the present analysis. Specifically, De Oliveira et al. [21] found that pediatric patients with CNS cancer had significantly higher pre-diagnosis costs over a 2-month period compared to leukemia. However, De Oliveira et al. [22] reported that among patients under 65 who survived beyond the first year after diagnosis, LC patients incurred the highest pre-diagnosis costs, followed by those with CRC, CNS cancer, and leukemia. Despite this, for patients who died within the first year of diagnosis, CNS cancer patients faced the highest costs, largely driven by in-patient care [22].

Some potential drivers of the cost difference across these cancers in the pre-diagnostic period include longer diagnostic intervals during which the accumulation of healthcare encounters leads to higher costs, the requirement for specialized tests and services due to the complexity or rarity of the diagnosis, or a combination of these factors [4]. Direct comparisons of diagnostic timing across these cancers are limited by the lack of data on diagnostic intervals in the literature. The stage at diagnosis is often used as a proxy for diagnostic timeliness and has been shown to be strongly associated with cancer outcomes [23]. However, this indicator is not applicable for CNS tumours, where severity is primarily assessed by grade rather than a progressive staging system [6,7,8]. Unlike many common cancers, CNS tumours do not necessarily advance to higher grades in the absence of treatment, complicating efforts to compare diagnostic timelines across cancer types included in this analysis. In Canada, overall survival rates for CNS cancers are similar to those for LC (an aggressive common cancer), yet pre-diagnostic hospital-based HEs are significantly higher for CNS cancers [24]. The similarity in survival across these cancers, despite differences in pre-diagnostic HE, may suggest that the higher costs observed for CNS cancers are driven more by the specialized tests and examinations required for these rare cancers than by longer diagnostic intervals. This interpretation aligns with the literature indicating that diagnosis of rare tumours may require more intricate diagnostic approaches once more common diseases are ruled out [4], increasing the costs associated with diagnosis. However, CNS cancers are highly heterogeneous, with 5-year net survival estimates ranging from 4.9% to 98.4%, so comparing overall outcomes between CNS cancers and other cancers may not fully capture the variation in diagnostic time, cost, and survival across different CNS cancer subtypes [25].

When comparing HE across provinces for the same cancer types, our findings indicated generally higher HE in Alberta, particularly for in-patient care across all age groups and cancer types. Conversely, out-patient care HE showed less variation across provinces. Given the hypothesis that increased costs might be associated with a longer diagnostic interval and, potentially, worse patient outcomes, we examined available surveillance estimates for the stage or grade at diagnosis and age-adjusted mortality across provinces. The data were available for CRC, LC, and CNS cancers. Surveillance data for colorectal and lung cancers across Alberta and Ontario showed no significant variation in the proportion of cancers diagnosed at later stages, suggesting that diagnostic intervals are not markedly different between these provinces [26,27]. For CNS tumours, we compared the distribution of tumour grades across provinces because tumour severity could influence diagnostic costs, given the specialized tests required for accurate diagnosis, and found no significant variation between Alberta and Ontario [25]. Similarly, age-adjusted mortality rates for these cancers do not significantly differ across provinces. This also suggests that the cost differences across provinces may reflect differences in diagnostic practices and patient management, which could be investigated to identify potential cost-saving measures. These findings emphasize the influence of regional variations in healthcare delivery within Canada’s publicly funded yet provincially administered healthcare system, where differences in care approaches can lead to substantial cost disparities [28]. Further investigation into these variations could uncover best practices that promote both cost-effectiveness and consistent quality of care across provinces.

Although our study identified some patterns indicating that patients with CNS incur higher pre-diagnostic costs compared to patients with common cancers, several limitations need to be considered, as they may impact the validity of these findings. First, our data reflected the healthcare expenditure from 10 years ago. Given the potential price inflation over the years, the recorded costs likely underestimate current expenditure. Further, our HE calculation only approximates the actual pre-diagnostic care cost as the exact diagnostic interval is unknown and not available in the literature, which limits our understanding of the actual cost. Second, some HE records were excluded during data extraction due to missing data, with the proportion of exclusions varying across cancer types. This could introduce bias in the comparisons due to differential selection. Third, the databases we used only provided total costs per patient record without detailing specific services or tests received. This limitation prevents us from identifying which particular interventions contributed to the overall costs, limiting our understanding of how different procedures could impact HE. Additionally, the data lacked information such as specific histological types, cancer stages, diagnostic procedures, and length of hospital stay. These factors are essential for a comprehensive analysis of HE, as they can significantly influence the costs incurred by patients. The absence of this information limits our understanding of the factors driving HE and the variation in costs among different cancer types. Lastly, the comparison group does not consist of patients with other types of common cancer. A descriptive study by de Oliveira et al. [22] provided a brief overview of pre-diagnosis costs for common cancers in ON. Based on their findings, the costs associated with the common cancers we selected were higher than the mean overall pre-diagnosis costs. This could limit the generalizability of our findings to a broader patient population. These limitations should be considered when interpreting the results of our study, and future research could address these issues to provide a more accurate assessment of HE associated with different types of cancer.

## 5. Conclusions

Overall, the findings from this analysis indicate that the diagnosis of CNS cancers incurs higher healthcare costs compared to common cancers, with the disparity becoming more pronounced at higher levels of expenditure. Additionally, these costs vary by province and patient encounter type. This work addresses a notable gap in the literature, as direct comparisons of diagnostic costs between rare and common cancers are limited. Although this analysis faced some limitations due to the availability of overall costs rather than detailed breakdowns of procedures, and limited details on diagnoses, it highlights critical areas for further investigation regarding variations in costs across different cancer types and provinces. Future research should aim to delve deeper into the specific factors driving these cost differences, including variations in diagnostic intervals, healthcare services received, differences in cancer staging, and histological subtypes. Additionally, future studies could explore the long-term economic impacts of rare cancers, like CNS cancers within the healthcare system, to inform more effective resource allocation and improve care strategies for patients with rare cancers, ultimately contributing to more equitable healthcare outcomes.

## Figures and Tables

**Figure 1 curroncol-32-00096-f001:**
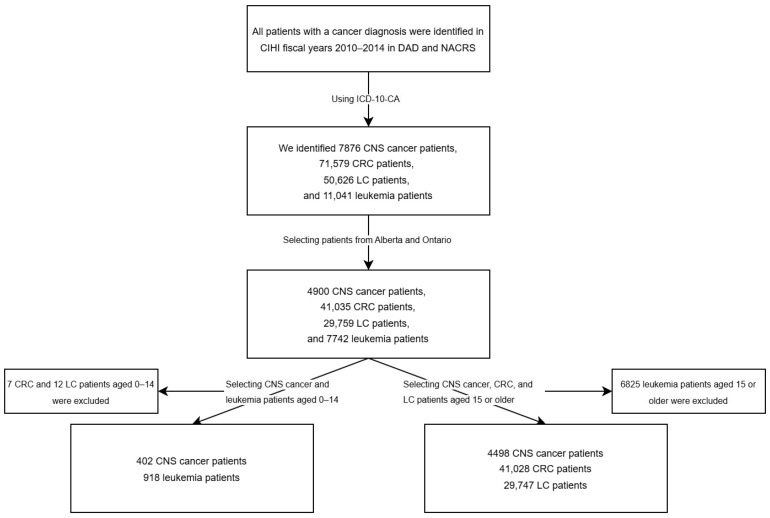
Flowchart of patient selection for this study.

**Figure 2 curroncol-32-00096-f002:**
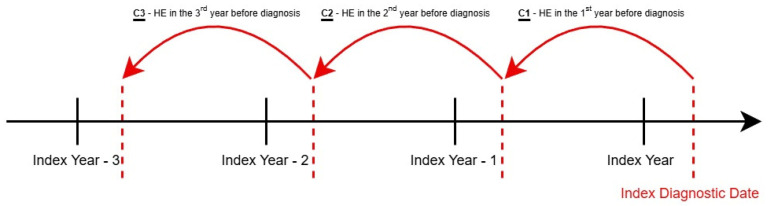
Pre-diagnostic healthcare expenditures (HEs) calculation method. Pre-diagnostic health care expenditure (HE) = $C1-\frac{\left(C2+C3\right)}{2}$.

**Figure 3 curroncol-32-00096-f003:**
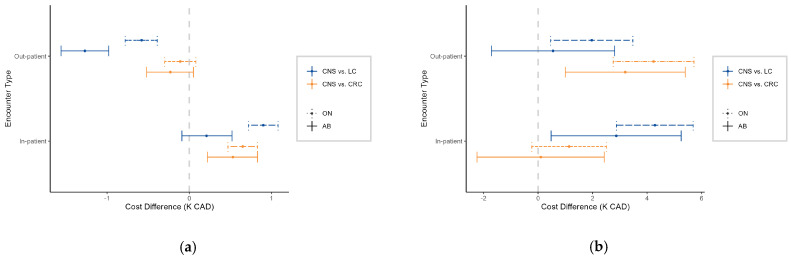
Cost difference (K CAD) between patients (aged 15 years and older) with central nervous system (CNS) and colorectal or lung cancer, stratified by province and encounter type at the (**a**) 50th percentile and (**b**) 90th percentile. Positive values indicate higher costs in CNS cancer patients.

**Figure 4 curroncol-32-00096-f004:**
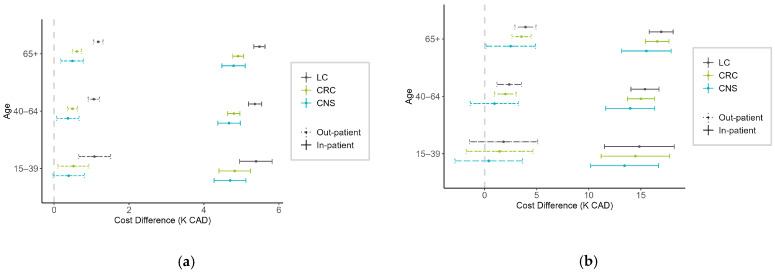
Cost difference (K CAD) between patients (aged 15 years and older) in Alberta and Ontario for different cancer types, stratified by age and encounter type at the (**a**) 50th percentile and (**b**) 90th percentile. Positive values indicate higher costs in Alberta.

**Figure 5 curroncol-32-00096-f005:**
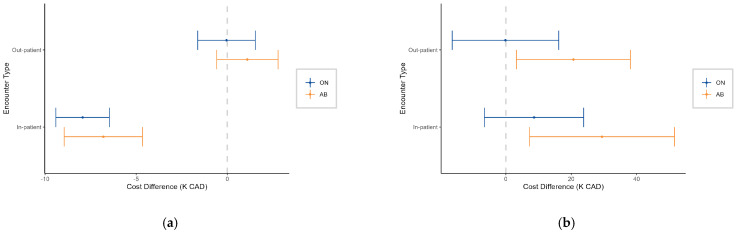
Cost difference (K CAD) between pediatric patients with central nervous system (CNS) and leukemia, stratified by province and encounter type at the (**a**) 50th percentile and (**b**) 90th percentile. Positive values indicate higher costs in CNS cancer patients.

**Figure 6 curroncol-32-00096-f006:**
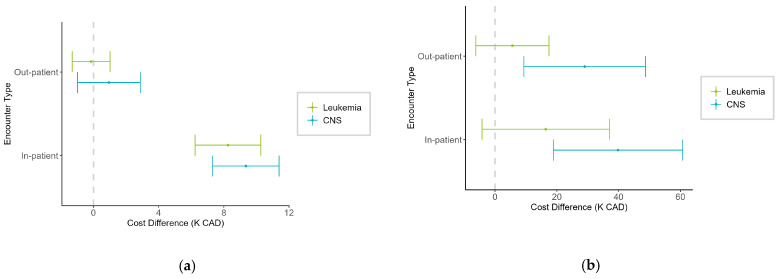
Cost difference (K CAD) between pediatric patients in Alberta and Ontario for different cancer types, stratified by encounter type at the (**a**) 50th percentile and (**b**) 90th percentile. Positive values indicate higher costs in Alberta.

**Table 1 curroncol-32-00096-t001:** Changes in total healthcare expenditure for patients (aged 15 years and older) with central nervous system tumours, colorectal cancer, or lung cancer in the ABON cohort using the DAD and NACRS (2010–2014) stratifying by patient characteristics: cost estimates at the 50th and 90th percentiles (in thousands of Canadian dollars).

	Central Nervous System			Colorectal			Lung		
	N (%)	Cost 50p	Cost 90p	N (%)	Cost 50p	Cost 90p	N (%)	Cost 50p	Cost 90p
Overall	4498	6.2	24.0	41,028	1.5	18.3	29,747	3.1	18.7
Province									
Ontario	3501 (78)	7.0	23.1	33,246 (81)	1.3	16.7	23,898 (80)	2.7	16.7
Alberta	997 (22)	3.9	25.6	7782 (19)	2.1	23.9	5849 (20)	4.5	25.5
Sex									
Female	2087 (46)	5.5	22.3	18,636 (45)	1.7	18.4	14,637 (49)	3.3	18.3
Male	2411 (54)	7.0	25.3	22,392 (55)	1.4	18.1	15,110 (51)	2.8	19.0
Age									
15–39 Years	829 (18)	8.1	21.2	865 (2)	2.0	14.9	209 (1)	2.7	21.6
40–64 Years	2036 (45)	5.6	22.1	14,071 (34)	1.2	13.5	9340 (31)	2.7	15.9
65+ Years	1633 (36)	6.1	27.1	26,094 (64)	1.7	21.4	20,198 (68)	3.3	20.0
Encounter Type									
In-patient	2454 (55)	11.0	31.5	13,905 (34)	11.1	35.1	11,289 (38)	11.0	30.5
Out-patient	2044 (45)	0.8	7.9	27,123 (66)	0.8	3.9	18,458 (62)	1.4	6.6
Comorbidity Status									
No	4204 (93)	7.6	24.7	36,232 (88)	1.7	19.1	24,469 (82)	4.3	20.3
Yes	294 (7)	0.8	6.4	4796 (12)	0.9	7.3	5278 (18)	1.4	7.2

**Table 2 curroncol-32-00096-t002:** Quantile regression for patients (aged 15 years and older) with central nervous system tumours, colorectal cancer, or lung cancer in the ABON cohort using the DAD and NACRS (2010–2014) at the 50th and 90th percentiles.

	At 50th Percentile		At 90th Percentile	
Characteristic	Coefficient (95% CI)	*p*-Value	Coefficient (95% CI)	*p*-Value
(Intercept)	11.4 (11.2, 11.6)	0.00	34.4 (33.0, 35.7)	0.00
Cancer type				
Central Nervous System	Ref.		Ref.	
Colorectal	−0.7 (−0.8, −0.5)	0.00	−1.1 (−2.5, 0.2)	0.10
Lung	−0.9 (−1.1, −0.7)	0.00	−4.3 (−5.7, −2.9)	0.00
Province				
Ontario	Ref.		Ref.	
Alberta	4.8 (4.5, 5.1)	0.00	15.5 (13.2, 17.9)	0.00
Age				
65+ Years	Ref.		Ref.	
15–39 Years	−1.1 (−1.4, −0.8)	0.00	−9.1 (−11.4, −6.9)	0.00
40–64 Years	−1.1 (−1.2, −1.0)	0.00	−9.0 (−9.8, −8.2)	0.00
Encounter Type				
In-patient	Ref.		Ref.	
Out-patient	−10.7 (−11.0, −10.4)	0.00	−26.1 (−28.0, −24.2)	0.00
Comorbidity				
No	Ref.		Ref.	
Yes	45.4 (45.0, 46.0)	0.00	109.2 (105.3, 113.1)	0.00
Cancer Type in Province				
Central Nervous System in Ontario	Ref.		Ref.	
Colorectal in Alberta	0.1 (−0.2, 0.4)	0.42	1.0 (−1.3, 3.4)	0.38
Lung in Alberta	0.7 (0.4, 1.0)	0.00	1.4 (−1.0, 3.8)	0.24
Cancer Type in Encounter Type				
Central Nervous System in In-patient	Ref.		Ref.	
Colorectal in Out-patient	0.8 (0.5, 1.0)	0.00	−3.1 (−5.0, −1.2)	0.00
Lung in Out-patient	1.5 (1.2, 1.7)	0.00	2.3 (0.4, 4.3)	0.02
Age in Province				
65+ Years in Ontario	Ref.		Ref.	
15–39 Years in Alberta	−0.1 (−0.5, 0.3)	0.65	−2.1 (−5.3, 1.1)	0.20
40–64 Years in Alberta	−0.1 (−0.3, 0.03)	0.11	−1.6 (−2.7, −0.4)	0.01
Encounter Type in Province				
In-patient in Ontario	Ref.		Ref.	
Out-patient in Alberta	−4.3 (−4.4, −4.2)	0.00	−13.0 (−14.2, −12.0)	0.00
Age in Encounter Type				
65+ Years in In-patient	Ref.		Ref.	
15–39 Years in Out-patient	1.1 (0.7, 1.5)	0.00	9.0 (6.2, 11.9)	0.00
40–64 Years in Out-patient	1.1 (1.0, 1.2)	0.00	7.2 (6.2, 8.1)	0.00
Comorbidity in Encounter Type				
No in In-patient	Ref.		Ref.	
Yes in Out-patient	−45.4 (−46.0, −44.9)	0.00	−108.7 (−112.6, −104.7)	0.00

**Table 3 curroncol-32-00096-t003:** Changes in total healthcare expenditure for pediatric patients with central nervous system tumours or leukemia in the ABON cohort using the DAD and NACRS (2010–2014), stratifying by patient characteristics: cost estimates at the 50th and 90th percentiles (in thousands of Canadian dollars).

	Central Nervous System			Leukemia		
	N (%)	Cost 50p	Cost 90p	N (%)	Cost 50p	Cost 90p
Overall	402	5.2	58.5	918	0.7	28.7
Province						
Ontario	267 (66)	5.9	41.2	593 (65)	0.8	28.7
Alberta	135 (34)	3.6	68.0	325 (35)	0.7	28.2
Sex						
Female	178 (44)	5.3	51.6	400 (44)	0.7	30.0
Male	224 (56)	4.2	72.5	518 (56)	0.7	27.5
Encounter Type						
In-patient	217 (54)	15.6	80.7	254 (28)	24.4	57.1
Out-patient	185 (46)	0.8	6.0	664 (72)	0.5	2.2
Comorbidity Status						
No	395 (98)	5.3	58.4	908 (99)	0.7	28.7
Yes	7 (2)	2.1	54.8	10 (1)	0.7	8.3

**Table 4 curroncol-32-00096-t004:** Quantile regression for pediatric patients with central nervous system (CNS) tumours or leukemia in the ABON cohort using the DAD and NACRS (2010–2014) at the 50th and 90th percentiles.

Characteristic	At 50th Percentile		At 90th Percentile	
	Coefficient (95% CI)	*p*-Value	Coefficient (95% CI)	*p*-Value
(Intercept)	14.3 (13.2, 15.5)	0.00	68.5 (55.9, 81.0)	0.00
Cancer Type				
Central Nervous System	Ref.		Ref.	
Leukemia	7.7 (6.2, 9.2)	0.00	−10.8 (−27.2, 5.6)	0.20
Province				
Ontario	Ref.		Ref.	
Alberta	9.6 (7.5, 11.6)	0.00	37.7 (15.1, 60.3)	0.00
Encounter Type				
In-patient	Ref.		Ref.	
Out-patient	−13.7 (−15.5, −12.0)	0.00	−67.0 (−86.3, −47.8)	0.00
Cancer Type in Province				
Central Nervous System in Ontario	Ref.		Ref.	
Leukemia in Alberta	−0.9 (−2.9, 1.2)	0.41	−18.5 (−41.3, 4.4)	0.11
Cancer Type in Encounter Type				
Central Nervous System in In-patient	Ref.		Ref.	
Leukemia in Out-patient	−7.8 (−9.7, −5.8)	0.00	10.8 (−11.0, 32.7)	0.33
Encounter Type in Province				
In-patient in Ontario	Ref.		Ref.	
Out-patient in Alberta	−8.8 (−10.9, −6.6)	0.00	−12.5 (−35.9, 11.0)	0.30

## Data Availability

Discharge Abstract Database (DAD) and National Ambulatory Care Reporting System (NACRS) metadata are available upon request from the Canadian Institute for Health Information.

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
