# Peer review of "Understanding Diagnostic Costs Using Hospital-Based Encounters in the Year Before Diagnosis for Canadian Patients with Malignant Central Nervous System Tumours Compared to Common Cancers"

_curroncol, 2025, doi:10.3390/curroncol32020096_

Round 1
Reviewer 1 Report
Comments and Suggestions for Authors
This study aims to establish the determinants of expenditure of rare tumors which, as is known, have higher healthcare costs than other tumors.
1. The authors correctly indicate in lines 36-40 some of the reasons that cause the higher cost of diagnosing rare cancers compared to common ones. The authors could consider the fact that the diagnostic methods recently developed for the diagnosis of rare tumors do not have low costs because it is not possible to achieve economies of scale, given the low number of tests required by the market.
2. The method used is clearly explained. The statistical methodology is appropriate.
3. At lines 268-271 the authors correctly indicate some of the reasons that may lead to an increase in diagnostic expenditure for rare cancers. They suggest that the difference between provinces depends on different levels of efficiency of regional services. The availability of diagnostic tests may explain the difference in costs between inpatient and outpatient settings. The cost resulting from comorbidity remains to be considered. On this topic, the authors could find literature.
4. The authors have indicated the limitations of the study. They could also provide some indications on the possible developments of the research and the applications of their results.
Reviewer 2 Report
Comments and Suggestions for Authors
Authors examined difference of healthcare expenditures of patients with malignant CNS tumours from those with colorectal cancer, lung cancer, and leukemia. However, this article has not fully answered some of the questions due to insufficient description.
First, authors suggest “Data were provided by the Canadian Institute for Health Information (CIHI) from two databases … we focused on patients from Alberta and Ontario (ABON) because these provinces have the most complete datasets, which covered patient data on emergency department visits, outpatient clinic visits, and day surgeries.” (L65), but it is difficult to understand how to select patients of this study. Authors should add flow chart to explain how to select patients of this study from those in databases.
Second, authors suggest “diagnosis codes were used to identify patients diagnosed with malignant CNS tumours during the fiscal years 2010 to 2014” (L78), but the months in a fiscal year vary from country to country. Authors should add month of fiscal years.
Third, authors suggest “Common cancers were selected based on annual surveillance estimates, including colorectal cancer (CRC), lung cancer (LC), and leukemia.” (L79), but they do not definition of “common cancers”, and they do not explain how to define them as common cancers. Without definition, it is difficult to justify what authors did. Authors should add explanation of why the selected colorectal cancer, lung cancer, and leukemia.
Fourth, authors suggest “From this index date, we traced back one to three fiscal years to extract all linkable historical records and estimate the cumulative pre-diagnostic costs for patients diagnosed with the cancers of interest each year. Pre-diagnostic healthcare expenditures (HE) were calculated as the change in total HE during the year before cancer diagnosis, determined by subtracting the average annual HE from the two years preceding the diagnosis from the HE in the year prior to diagnosis.” (L92), but it is difficult to understand how to calculate pre-diagnostic healthcare expenditures. Authors should add figure to explain how to calculate pre-diagnostic healthcare expenditures.
Fifth, authors do not show 95% confidence intervals for some of variable in Table 2 and Table 3. Author may suggest “We included variables and scientifically relevant interactions in the final regression model if their p-value was below 0.2” (L148), but some of variables with >0.2 of p-value have 95% confidence intervals. Authors should add 95% confidence intervals as well as p-value for all variable in Table 2 and Table 3.
Finally, authors described some of sentences without citation or justification as follows; “This group of rare tumours is characterized by a wide range of histopathological subtypes, each associated with different treatment requirements and prognoses, making accurate classification essential for optimal patient management.” (L46), “As a result, the diagnostic interval for CNS tumours can be extended not only by the time required to rule out more common explanations for early symptoms but also by the need for specialized diagnostic tests and the expertise necessary to select the appropriate tests.” (L55), and “many studies have explored the economic burden of cancer” (L258), but it is difficult for readers to judge them without references as evidence for each description. Authors should add references for these descriptions.
Round 2
Reviewer 1 Report
Comments and Suggestions for Authors
The authors revised the manuscript according to suggestions
Author Response
Comment 1: The authors revised the manuscript according to suggestions
Response 1: We sincerely thank you for taking the time to review our manuscript and for your valuable comments and suggestions.
Reviewer 2 Report
Comments and Suggestions for Authors
Authors revised the manuscript, but this article has not fully answered some of the questions due to insufficient description.
First, authors add a flow chart as Figure 1, but the total numbers of patients aged 0-14 and those aged 15 or older do not add up for CRC, LC, and leukemia, which means some patients may be excluded from the patients selected from Alberta and Ontario. Authors should check the number of patients, carefully.
Second, as mentioned in the previous review, authors do not definition of “common cancers”, and they do not explain how to define them as common cancers in the manuscript. Authors suggest “we have provided definitions for rare cancers instead, as follows in L31”, but it is difficult for readers to justify what authors did without definition in the manuscript. Authors should add explanation of why the selected colorectal cancer, lung cancer, and leukemia as “common cancers” in the manuscript.
Third, as mentioned in the previous review, authors do not show 95% confidence intervals for some of variable in Table 2 and Table 4. Authors suggest “For Tables 2 and 4, we included all possible interaction terms in the models. As some combinations had neither effects nor confidence intervals, we replaced these with a dash (–) to indicate NA combinations and to reduce confusion.”, but it is difficult for readers to justify what authors did without descriptions of the results in the manuscript. Authors should add estimations and 95% confidence intervals as well as p-value for all variable in Table 2 and Table 4.
Finally, authors do not show differences using statistical results (e.g., p-values) in Table 1 and Table 3. Authors suggest “Tables 1 and 3 were used to provide a summary to the data; we did not calculate p-values as these tables were not used to make inferences.”, but authors compare costs by some group in result section (e.g., “This pattern persisted at the 90th percentile, with CNS patients costs being $5.7K higher than CRC and $5.3K higher than LC patients. From the provincial perspective, HE was generally higher in Alberta than Ontario for all cancer types.” (L133)) in the result section. It is difficult for readers to justify what authors did without results of statistical analyses in the manuscript. Authors should add results of statistical analyses (e.g. p-values) in Table 1 and Table 3.
Round 3
Reviewer 2 Report
Comments and Suggestions for Authors
Authors revised the manuscript, but this article has not fully answered some of the questions due to insufficient description.
First, as mentioned in the previous review, the total numbers of patients aged 0-14 and those aged 15 or older do not add up for CRC, LC, and leukemia, which means some patients may be excluded from the patients selected from Alberta and Ontario. Authors suggest “For CRC, LC and leukemia patients, only a subset met our inclusion criteria based on age and were analyzed”, but if so, they should add another box for the excluded participants in Figure 1. Authors should revise Figure 1.
Second, as mentioned in the previous review, authors do not show differences using statistical results (e.g., p-values) in Table 1 and Table 3. Authors suggest “Our intention was to describe the population, or present the distribution of characteristics in the population. We did not perform statistical comparisons between groups; instead, we focused on describing the observed patterns.”, but authors compare costs by some group in result section (e.g., “This pattern persisted at the 90th percentile, with CNS patients costs being $5.7K higher than CRC and $5.3K higher than LC patients.” (L135)) in the result section. It is difficult for readers to justify what authors did without results of statistical analyses in the manuscript. Authors should add results of statistical analyses (e.g. p-values) in Table 1 and Table 3.
